# Therapeutic potential of salidroside in type I diabetic erectile dysfunction: Attenuation of oxidative stress and apoptosis via the Nrf2/HO-1 pathway

Zhenghao Li[1], Bin Jia[2], Zhongkai Guo[1], Keqin Zhang[2], Danfeng Zhao[2], Ziheng Li[3], Qiang Fu [1,2,4,5]*

1 Department of Urology, Shandong Provincial Hospital, Shandong University, Jinan, Shandong, China, 2 Department of Urology, Shandong Provincial Hospital Affiliated to Shandong First Medical University, Jinan, Shandong, China, 3 Second Department of Surgery, Shandong Rongjun General Hospital, Jinan, Shandong, China, 4 Key Laboratory of Urinary Diseases in Universities of Shandong (Shandong First Medical University), Jinan, Shandong, China, 5 Engineering Laboratory of Urinary Organ and Functional Reconstruction of Shandong Province, Shandong Provincial Hospital Affiliated to Shandong First Medical University, Jinan, Shandong, China

* qiangfu68@163.com

**Data Availability Statement:** All relevant data are within the manuscript and its Supporting Information files.

## Abstract

The primary objective of this work was to delve into the potential therapeutic advantages and dissect the molecular mechanisms of salidroside in enhancing erectile function in rats afflicted with diabetic microvascular erectile dysfunction (DMED), addressing both the whole-animal and cellular dimensions. We established a DMED model in Sprague–Dawley (SD) rats and conducted in vivo experiments. The DMED rats were administered varying doses of salidroside, the effects of which on DMED were compared. Erectile function was evaluated by applying electrical stimulation to the cavernous nerves and measuring intracavernous pressure in real time. The penile tissue underwent histological examination and Western blotting. Hydrogen peroxide ($H_2O_2$) was employed in the in vitro trial to induce an oxidative stress for the purpose of identifying alterations in cell viability. The CCK-8 assay was used to measure the viability of corpus cavernous smooth muscle cells (CCSMCs) treated with vs. without salidroside. Flow cytometry was utilized to detect alterations in intracellular reactive oxygen species (ROS). Apoptosis was assessed through Western blotting and TdT-mediated dUTP nick-end labelling (TUNEL). Animal and cellular experiments indicate that the Nrf2/HO-1 signalling pathway may be upregulated by salidroside, leading to the improvement of erectile function in diabetic male rats by alleviating oxidative stress and reducing apoptosis in corpus cavernosum tissue.

## Introduction

Erectile dysfunction (ED) is a clinical state distinguished by the consistent or recurrent incapability to attain and/or sustain a satisfactory erection to achieve sexual contentment. Common

**Funding:** The author(s) received no specific funding for this work.

**Competing interests:** The authors have declared that no competing interests exist.

**Abbreviations:** DM, Diabetes mellitus; ED, Erectile dysfunction; DMED, Diabetic mellitus erectile dysfunction; STZ, Streptozotocin; PDE5i, Phosphodiesterase type 5 inhibitors; Nrf2, Nuclear factor erythroid-2 related factor 2; HO-1, Haem oxygenase-1; TUNEL, TdT-mediated dUTP Nick-End Labelling; Bcl-2, B-cell lymphoma-2; Bax, Bcl-2 associated X Protein; Cleaved-caspase-3, Cleaved-cysteine aspartic acid specific protease-3; α-SMA, α-Smooth muscle actin; vWF, Von Willebrand factor; eNOS, Endothelial Nitric Oxide Synthase; ICP, Intracavernous pressure; Max ICP, Maximal intracavernous pressure; MAP, Mean arterial blood pressure; CN, Cavernous nerve; ROS, Reactive oxygen species; MDA, Malondialdehyde; SOD, Superoxide Dismutase; CCSMCs, Corpus cavernous smooth muscle cells.

factors raising the likelihood of ED are advancing age, diabetes, dyslipidaemia, hypertension, cardiac ailments, obesity, metabolic syndrome, a sedentary way of life, and tobacco usage [1,2].

In individuals afflicted by diabetes mellitus (DM), elevated concentrations of glucose in the bloodstream are linked to heightened oxidative stress and excessive production of reactive oxygen species (ROS). This triggers a cascade of occurrences within the context of DM, including a decline in nitric oxide (NO) levels, a rise in prothrombotic factors (such as tissue factors and plasminogen activator inhibitor-1), a rise in endothelin-1, and episodes of thrombosis and vasoconstriction. Moreover, nuclear factor kappa B and activating protein 1 are upregulated, giving rise to inflammation and ultimately culminating in diabetic microvascular endothelial dysfunction (DMED) [3].

Nuclear factor erythroid-2 related factor 2 (Nrf2) has a crucial role in the regulation of oxidative stress through both antioxidant and anti-inflammatory activity. This specific protein stimulates more than 50 genes associated with oxidation–reduction reactions and almost 200 genes related to the processes of metabolism and restoration [4]. Nrf2 plays a crucial role in safeguarding cells and aids in fighting stress caused by the environment [5]. Haem oxygenase-1 (HO-1) acts as the enzyme that controls the speed of haem degradation. It facilitates the conversion of harmful haem into iron ions, carbon monoxide, and biliverdin, which is later transformed into bilirubin [6]. The Nrf2/HO-1 signalling pathway can alleviate oxidative harm, control cell apoptosis, adjust inflammation, and enhance vascular formation, supporting physiological well-being. The Nrf2/HO-1 signalling pathway is upregulated during acute hyperglycaemia but decline during chronic and persistent hyperglycaemia, suggesting a potential involvement of this pathway in activating the antioxidant system during stressful circumstances [7]. Nevertheless, excessive stress can cause a reduction in Nrf2 and HO-1 levels, leading to a diminished ability to combat oxidative stress and an escalation of oxidative harm, which contribute to the emergence of diabetic complications such as DMED [8].

Cysteine aspartic acid-specific protease-3 (caspase-3), an enzyme that specifically targets its cysteine to the substrate's aspartic acid, plays a crucial role in cellular apoptosis. When apoptotic signals such as oxidative stress, DNA damage, cytotoxic drugs, and viral infection factors are received, caspase-3 undergoes cleavage into two active subunits known as cleaved caspase-3, which subsequently triggers cellular apoptosis. The B-cell lymphoma-2 (Bcl-2) gene family plays a crucial role in controlling apoptotic signal transduction [9]. This family consists of anti-apoptotic proteins such as Bcl-2 and proapoptotic proteins such as Bcl-2-associated X protein (Bax). In cases of erectile dysfunction in individuals with diabetes, caspase-3 and Bax are upregulated and Bcl-2 is downregulated in the tissue of the corpus cavernosum in rats. This ultimately results in an augmented occurrence of apoptosis in the smooth muscle cells of the cavernous region (CSMCs) [10]. Comparable alterations have been noted in CCSMCs cultivated in vitro under elevated glucose circumstances, indicating that programmed cell death plays a significant part in the progression of diabetic impotence [11].

Salidroside is a bioactive compound found in the plant *Rhodiola rosea*. Also known as golden root or Arctic Rhodiola, this is a perennial plant that blooms and thrives in frigid regions such as the Arctic and mountainous parts of Europe, Asia, and North America [12]. The botanical has been utilized for many years in traditional healing practices, such as traditional Chinese medicine (TCM) and Ayurveda, because of its alleged adaptogenic and fatigue-fighting characteristics [13]. The potential therapeutic properties of salidroside have garnered considerable interest from scientists [14]. Several studies have indicated that salidroside has a broad spectrum of advantageous effects, encompassing antioxidant, anti-inflammatory, neuroprotective, cardioprotective, and anticancer properties [15–18]. An imbalance between the generation of ROS and the body's antioxidant defence system, called oxidative stress, has been linked to numerous ailments, such as cardiovascular diseases, neurodegenerative disorders,

and cancer. Salidroside has demonstrated strong antioxidant properties, effectively removing ROS and protecting cells against oxidative harm [19]. In an effort to investigate its mechanisms of action, we administered salidroside orally to diabetic rats to assess its therapeutic impact on DMED.

## Methods

### Animals

All experimental procedures involving animal management were granted approval by the Animal Care and Use Committee at the University of Jinan, China. A combined number of 36 male Sprague–Dawley rats, with an age of 8 weeks and a weight ranging from 260 to 300 g, were provided by the Animal Center of Shandong University, China. The rats were accommodated in a controlled environment free from pathogenic microorganisms, with a constant temperature of $23 \pm 1°C$ and a 12-hour light-dark cycle. Throughout the research process, they were provided unrestricted opportunities to food and water. The animal care and treatment procedures followed the guidelines established by the Animal Care and Use Committee of Shandong University (Jinan, China). To induce diabetes, the rats underwent a 12-hour fast, after which they received a solitary intraperitoneal injection of 55 mg/kg streptozotocin (STZ, Sigma–Aldrich). Their blood glucose was randomly monitored within 72 hours after STZ injection. Rats exhibiting a spontaneous blood glucose > 16.7 mmol/L were classified as diabetic. After 8 weeks, DM rats were administered a subcutaneous injection of 100 μg/kg apomorphine (APO, Sigma–Aldrich) in the posterior neck region. Following the injection, each rat was observed for 30 minutes for penile erection. If no erection occurred, the rat was classified as a DMED rat. A total of 27 DMED rats were screened. The rats with DMED were categorized into three distinct groups: (1) salidroside (Selleckchem, USA, IG) was given at a daily dose of 20 mg/kg/day (DM+LOW group, n = 9), (2) salidroside (IG) was given at a daily dose of 40 mg/kg/day (DM+HIGH group, n = 9), and (3) a control DMED group (DM group, n = 9) not given salidroside. Throughout the experiment, the blood glucose levels at the beginning and end, along with the body weights, were documented for all the rats. In subsequent experiments, 3% pentobarbital sodium was used for anaesthesia in order to reduce the pain of the experimental animals, and at the end of the experiments, the animals were put to death by deep anaesthesia.

### Evaluation of erectile function

To evaluate the functionality of the corpus cavernosum, rats from the different groups underwent evaluation of intracavernosal pressure (ICP) and mean systemic arterial pressure (MAP) following an eight-week treatment duration. After the administration of 3% pentobarbital sodium, the male rats were placed in a supine position. For the measurement of systemic arterial blood pressure, we employed a 25-gauge needle connected to a PE-50 tube filled with heparinized saline (300 IU ml-1). This needle was carefully inserted into the carotid artery on the left side to determine MAP. Additionally, we introduced another needle into the cavernous body to monitor ICP.The cavernous nerve (CN) was revealed through a surgical incision along the midline and stimulated using a hook-shaped stainless steel electrode with parameters: 25 Hz, a pulse width of 5 ms, 1.5 mA and a duration of 60 s. The BL-420V pressure transducer system (AD Instruments, Sydney, Australia) was utilized to record and visualize variations in both ICP and MAP.

The mice were put in a surgical chamber to acclimate to their surroundings for at least 20 minutes. Following anaesthesia, the rat was positioned on a steady-temperature platform. To prevent any harm to the skin and avoid biasing the results, the hair removal procedure was

conducted with great care. The FLIR T540 thermal imager, produced by FLIR Systems in Boston, MA, was placed about 25 cm from the rat's genital region. A temperature gauge was inserted into the rat's anus. The rat's erectile response was observed when CN stimulation or subcutaneous injection of APO was used to induce erectile stimulation [20].

## Masson's trichrome staining

Euthanasia was performed 30 minutes following the assessment of erectile function. An incision was made in the skin of the rat's penis to reveal the penile crura on both sides, and a section of the muscle and fascia of the penile corpus cavernosum was removed. The male reproductive organ was detached from the tip. The tissues of the penis were randomly divided into two sections of similar size and shape. A specific portion was employed for the separation of the corpus cavernosum and was examined by Western blot. Immunofluorescence staining and histological examination were conducted on the remaining portion of penile tissue. For the examination of smooth muscle cells and collagen fibrils within the corpus cavernosum, a Masson's trichrome staining kit (Dako Sciences, Glostrup, Denmark) was employed. Collagen fibres were specifically coloured blue, elastic fibres brown. Muscle fibres were stained with a red hue, and nuclei were subsequently counterstained in a deep blue colour. The quantification of the ratio between smooth muscle and collagen was performed utilizing Image-Pro Plus 6.0 software.

## Immunofluorescence staining

The cellular components and frozen tissue sections were immobilized using 4% paraformaldehyde and exposed to 0.5% Triton X-100 for approximately 20 minutes at ambient temperature. Next, the sections were subjected to an incubation step involving goat serum for 30 minutes. Each slide was overlaid with a primary antibody: anti-$\alpha$-smooth muscle actin ($\alpha$-SMA, 1:500; Servicebio), anti-von Willebrand factor (vWF, 1:500; Servicebio), or anti-endothelial nitric oxide synthase (eNOS, 1:500; Servicebio). After incubation at 4˚C overnight, the sections were treated with suitable secondary antibodies (Invitrogen, Carlsbad, CA, USA) for 1 hour at 37˚C. Finally, the slices were stained with DAPI (Invitrogen) and examined under a Nikon microscope (Nikon, Tokyo, Japan).

## TUNEL staining

For immunofluorescence staining, sections were prepared from the isolated corpus cavernosum tissue. Apoptotic cells were identified using an in situ apoptosis assay kit (Nanjing Novozan Biotechnology Co., Ltd.). Microscopic images were acquired using a Nikon Eclipse C1 microscope. Apoptotic cells were visualized through green fluorescence, contrasting with the blue fluorescence exhibited by the nuclei.

## Western blotting

Tissue and cellular extracts were formulated by combining RIPA buffer with a mixture of protease inhibitors. The BCA assay was utilized to determine the protein concentrations in the samples. Protein samples were each treated with 20 μg and subsequently underwent sodium dodecyl sulfate–polyacrylamide gel electrophoresis (SDS-PAGE) prior to being transferred onto a polyvinylidene fluoride (PVDF) membrane. The membrane underwent blockage by 5% skim milk and subsequently experienced overnight incubation at a temperature of 4˚C with primary antibodies. The primary antibodies were specifically directed towards Nrf2 (at a 1:2000 dilution, from Proteintech), HO-1 (at a 1:2000 dilution, from Proteintech), Bcl-2 (at a 1:2000 dilution, from Proteintech), Bax (at a 1:2000 dilution, from Proteintech), and cleaved

caspase-3 (at a 1:2000 dilution, from Proteintech).After incubation with the suitable secondary antibodies, the signal was observed using a LAS3000 Image Analyzer (Fujifilm, Tokyo, Japan), and the data were subsequently analysed utilizing Multigauge software (Fujifilm).

## The level of SOD enzyme activity and the amount of MDA in penis tissue

In order to evaluate the oxygenation status within the cavernous sinus, levels of malondialdehyde (MDA) and superoxide dismutase (SOD) were assessed. The MDA content in the cavernous body was determined as specified by the assay kit (Nanjing Jiancheng Biochemical Co., Ltd., Nanjing, China). The reaction between MDA and thiobarbituric acid produces a complex that is pink in colour and has a maximum absorbance wavelength of 532 nm [21]. The level of MDA is indicated in millimoles per milligram of protein. To measure the overall SOD activity, a spectrophotometric assay kit was utilized. It detects the highest absorbance wavelength at 525 nm. The findings are characterized as units per milligram of protein.

## Cell culture

As described [22], corpus cavernous smooth muscle cells(CCSMCs) were isolated from the cavernous bodies of young male SD rats, aged 8 weeks. Following the removal of the outer layer and inner lining, the spongy tissues were dissected and placed in containers for cultivation. The tissue samples were cultured in DMEM with low glucose, along with 10% foetal bovine serum and 1% penicillin–streptomycin, then transferred and refined for additional investigations. Staining with $\alpha$-smooth muscle actin ($\alpha$-SMA) (1:200, Servicebio) was conducted to verify the identity of CCSMCs. CCSMCs from stages IV to VIII were used in the experiments. Following a 24-hour incubation period using serum-deprived DMEM, the cells were categorized into three groups: Sham, $H_2O_2$ group ($H_2O_2$, 200 $\mu$mol/ml), and Sal group (salidroside, 40 $\mu$g/ml). The allocation of cell groups was set prior to data collection, following a similar approach used in the in vivo experiment.

## Cell viability was measured using CCK-8

The viability of CCSMCs was assessed using the Cell Counting Kit-8 (CCK-8) assay. For culturing CCSMCs, a 96-well plate was utilized, seeding 1000 cells per well. Afterward, adequate time was allotted for the cells to attach overnight. The cells were exposed to various levels of $H_2O_2$ (100, 200, 400, 600, and 800 $\mu$M) for different times (2, 4, 6, 12, and 24 hours). The ideal amount and duration of treatment for $H_2O_2$ were found to be 200 $\mu$M for 12 hours. The cells were categorized into three groups: sham group, $H_2O_2$ group, and salidroside group. Normal culture medium was used to treat the sham group. The cells in the salidroside group were cultured with 40 $\mu$g/ml salicylic acid for 12 hours. The salidroside group and the $H_2O_2$ group were subjected to $H_2O_2$ at a 200 $\mu$M concentration for an extra 12 hours. Normal culture medium was used to treat the sham group in this experiment. To determine the optical density values of the samples, a 10 $\mu$L aliquot of CCK-8 solution (MedChemExpress, New Jersey, USA) was added to each well at specific time intervals. The samples were then incubated for an additional 4 hours. After the incubation period, the optical density values at 450 nm were measured using a microplate reader (BioTek, Winooski, VT, USA). Every group underwent three tests, and the experiment was repeated on three occasions.

## Flow cytometry

We assessed the cellular levels of ROS using the CM-H2DCFDA cell-permeable probe (MedChemExpress, New Jersey, USA). This particular probe has the ability to effectively penetrate

the cellular environment and detect 5-(and-6)-chloromethyl-2',7'-dichlorodihydrofluorescein diacetate. To perform the analysis, the cells were incubated in DMEM (without phenol red) containing 10 μM H2DCFDA for a duration of one hour. Following the incubation period, the cells underwent two rounds of rinsing with DMEM. To evaluate the results, we employed a flow cytometer from ACEA (model: NovoCyte, originating from the USA) and subsequently analyzed the collected data using NovoExpress software.

## Statistical analysis

The experiments in this study were conducted multiple times to ensure accuracy and reliability. The results are presented with means and standard deviations to provide a comprehensive understanding of the data. The statistical analysis was performed using SPSS software, specifically version 22.0 from SPSS in Illinois, USA. To compare differences between two groups, Student's t test was used, while for comparisons between three groups, one-way analysis of variance (ANOVA) was employed. A significance level of $P < 0.05$ was considered statistically significant.

## Ethical approval

Approval for all animal-related procedures, encompassing animal treatment and handling, was granted by the Shandong Provincial Hospital's Institutional Animal Care and Use Committee.

## Results

### In diabetic rats, salidroside does not impact body weight or blood glucose

Table 1 displays the the weights of the bodies and the concentrations of blood glucose. Before conducting the experiment, it was observed that the diabetic rats exhibited considerably higher levels of blood glucose compared to the sham group. Additionally, these rats displayed a substantial reduction in weight, which was significantly more pronounced ($P < 0.01$). The administration of salidroside did not lead to any change in blood sugar, though it did lead to significant weight gain ($P < 0.01$).

### Salidroside enhance erectile function in diabetic rats

We assessed the alterations in rats' erectile function with various intervention methods, using the maximum ICP and the ratio of ICP to mean arterial pressure (MAP) as indicators (Fig 1A). Significant reductions in both the maximum ICP and the ICP/MAP ratio were

**Table 1. Weight and blood glucose in experimental animals.**

|  |  | Sham (n = 9) | DM (n = 9) | DM+L (n = 9) | DM+H (n = 9) |
|---|---|---|---|---|---|
| Initial | Weight (g) | 280.8±20.2 | 279.1±21.6 | 273.6±23.3 | 276.8±22.4 |
|  | Glucose (mmol/L) | 5.96±1.88 | 25.89±6.20 | 26.23±5.10 | 26.09±4.26 |
| Post-intervention | Weight (g) | 518.8±43.1 | 348.1±38.5** | 386.9±45.6## | 443.6±40.7## |
|  | Glucose (mmol/L) | 6.13±1.45 | 26.51±5.19** | 25.62±3.97# | 23.09±3.52## |

Data are displayed as mean ± standard deviation for each group (n = 9).

**P < 0.01 indicates significant distinctions in comparison to the sham group.

#P < 0.05 and

##P < 0.01 indicate significant disparities in comparison to the DM group.

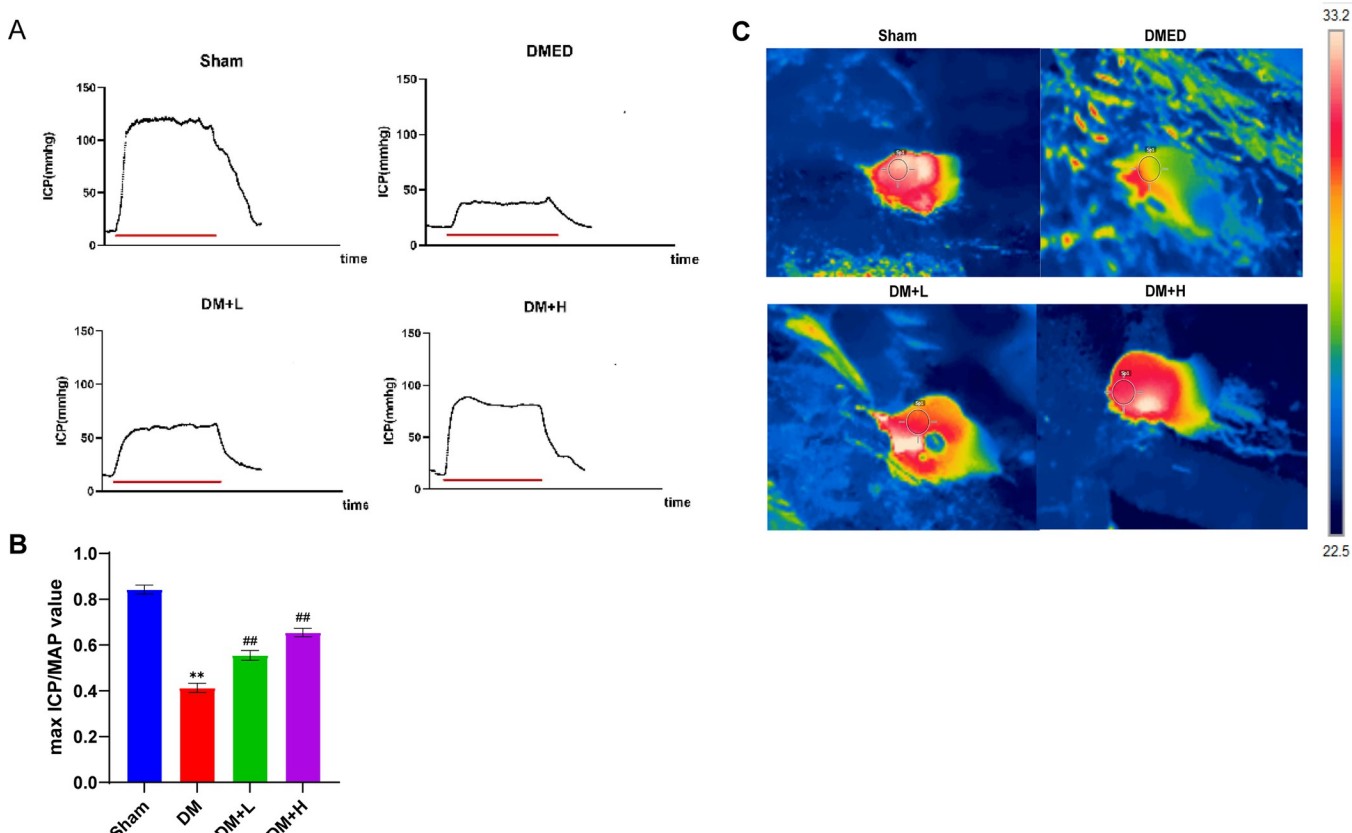

**Fig 1. Changes in erectile function in each group of rats.** (A and B) Salidroside therapy enhanced erectile dysfunction induced by electrical activation of the CN. Crimson bars represent 60 seconds of CN electrical stimulation; the proportion of the highest ICP to MAP in all three groups is illustrated as a bar chart of the highest ICP/MAP. Data are presented as mean±SD, n = 9. **P<0.01 signifies a noteworthy distinction compared to the sham group. ##P<0.01 signifies a noteworthy distinction compared to the DM group. (C) Real-time thermal imaging observation of penile engorgement during erection in each group of rats.

observed in the DM group of rats when compared to the normal rats (P<0.01, Fig 1B). Compared to diabetic rats, rats administered with a low dose of salidroside showed some improvement in erectile dysfunction, whereas rats administered with a high dose of salidroside exhibited significant improvement (P<0.01, Fig 1B). Furthermore, rats in the DM group exhibited a notable decline in blood circulation to the penile vessels during the state of erection, which was significantly improved after salidroside treatment (Fig 1C).

## Salidroside attenuates penile corpus cavernosum fibrosis and apoptosis in DM rats

The ratio of corpus cavernosum smooth-muscle-to-collagen was quantified by slides stained with Masson's trichrome. Smooth muscle content was restored by treatment with both low and high doses of salidroside, as depicted in Fig 2A. Fig 2C displays the ratios of collagen to smooth muscle for each group. The salidroside-treated groups had higher smooth muscle density (P<0.01) than the DM group. The TUNEL assay revealed that salidroside treatment resulted in a notable reduction in apoptosis of corpus cavernosum tissue cells in DMED rats. Particularly, the high dose of salidroside demonstrated a more notable reduction in cellular apoptosis (Fig 2D).

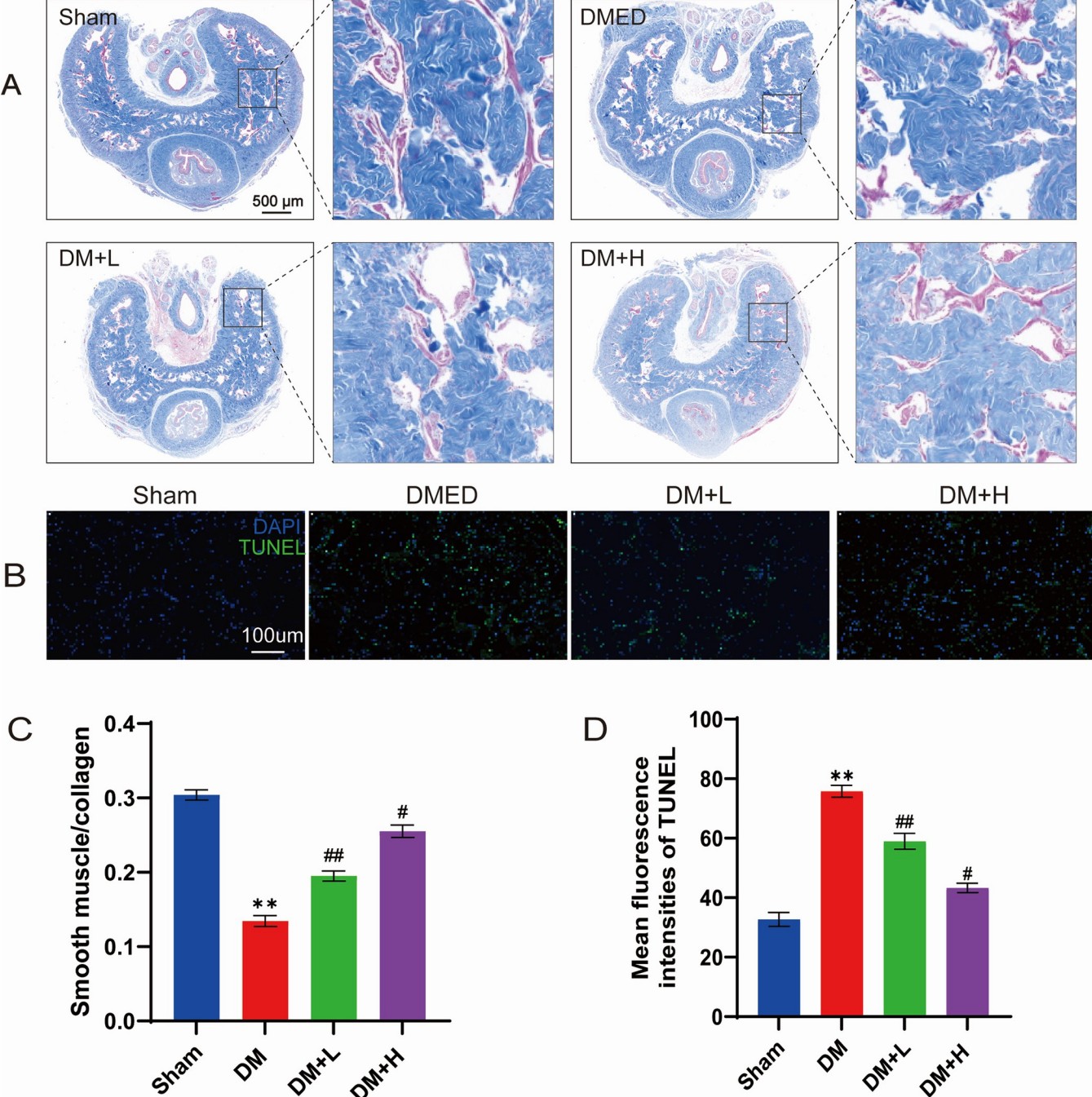

**Fig 2. Salidroside alleviates fibrosis and apoptosis of the cavernous body of the penis in DMED rats.** (A) Masson's trichrome staining indicated the level of fibrosis in the cavernous bodies of rats from all four experimental groups following various treatments (red represents the smooth muscle area, while blue represents the collagen area). (B) Apoptosis of rat corpus cavernosum determined by TUNEL assay. (C and D) We utilized the Image-Pro Plus software to quantify the proportion of collagen to smooth muscle, while also measuring the mean fluorescence intensities of the TUNEL assay. The analysis of the data was conducted using the GraphPad Prism 9 software, and the results are displayed as the mean ± SD. To evaluate the differences between two groups, we employed an unpaired t-test. For comparisons among three groups, a one-way ANOVA followed by Tukey's post hoc test was performed. \*\*$P<0.01$ signifies a noteworthy distinction compared to the sham group. ##$P<0.01$ #$P<0.05$ signifies a noteworthy distinction compared to the DM group.

### The function of endothelial cells is enhanced by salidroside

Analysis of α-SMA expression highlighted a significant decrease in smooth muscle content in diabetic rats through immunofluorescence staining. In contrast to the sham group, the DM group exhibited reduced levels of von Willebrand factor (vWF) and endothelial nitric oxide synthase (eNOS) within the endothelium (Fig 3A–3D). Under salidroside treatment, the endothelial cell content increased.

### Measurement of SOD function and MDA concentration in the tissue of the corpus cavernosum of rats

The antioxidative metal enzyme SOD plays a vital part in maintaining the equilibrium between oxidation and antioxidation in the body, carries out the conversion of superoxide radicals into both oxygen and hydrogen peroxide, thus contributing to the process of antioxidation. The degree of lipid peroxidation and cellular damage can be determined by the content of MDA, which is the result of lipid peroxidation. To evaluate the oxidative stress condition of the rat corpus cavernosum, the levels of SOD activity and MDA were measured. Rats in the DMED group had lower SOD activity ($P<0.01$, Fig 3E) and higher MDA content than the Sham group ($P<0.01$, Fig 3F), suggesting an enhanced state of oxidative stress. In comparison to the DM group, rats that received low and high doses of salidroside exhibited elevated SOD activity ($P < 0.01$, $P < 0.05$) and reduced MDA content ($P < 0.01$) in the corpus cavernosum tissue. The high-dose treatment group with salidroside exhibited a more noticeable therapeutic impact.

### Salidroside activates the Nrf2/HO-1 antioxidant stress pathway

Fig 4A shows that the corpus cavernosum of DM rats had downregulated Nrf2 and HO-1 expression, as indicated by Western blotting ($P < 0.01$, Fig 4B and 4C). The intervention group that received salidroside treatment showed a notable rise in the levels of Nrf2 and HO-1 compared to the DM group ($P < 0.01$), indicating a correlation that varied based on dosage.

### Salidroside reduces apoptosis in diabetic penile corpus cavernosum

The Western blotting findings revealed a notable decrease ($P<0.01$, Fig 4A) in Bcl-2 expression, indicating a diminished antiapoptotic capacity, in the corpus cavernosum tissue of diabetic rats. Conversely, the expression levels of Bax and cleaved caspase-3 were significantly increased, suggesting a higher rate of apoptosis (Fig 4A, $p<0.01$). In contrast, the penile tissue from the low and high-dosage salidroside treatment groups exhibited increased Bcl-2 expression (Fig 4D, $p<0.01$) and decreased expression of Bax and cleaved caspase-3 (Fig 4E and 4F, $p<0.01$) compared to the diabetic group.

### Isolation and characterization of penile CCSMCs

CCSMCs displayed an elongated and spindle-shaped morphology (Fig 5A). The immunofluorescence staining analysis unraveled that almost all ($<95\%$) of the cells exhibited a strong expression of the smooth muscle cell marker α-SMA, indicating a high purity of the cultured primary cells (Fig 5B).

### The viability of CCSMCs is improved by pretreatment with salidroside when exposed to H2O2

We assessed the viability of CCSMCs under H2O2 intervention by performing CCK-8 assays. The findings indicated a notable decline in cell viability in CCSMCs exposed to 200 μmol/L

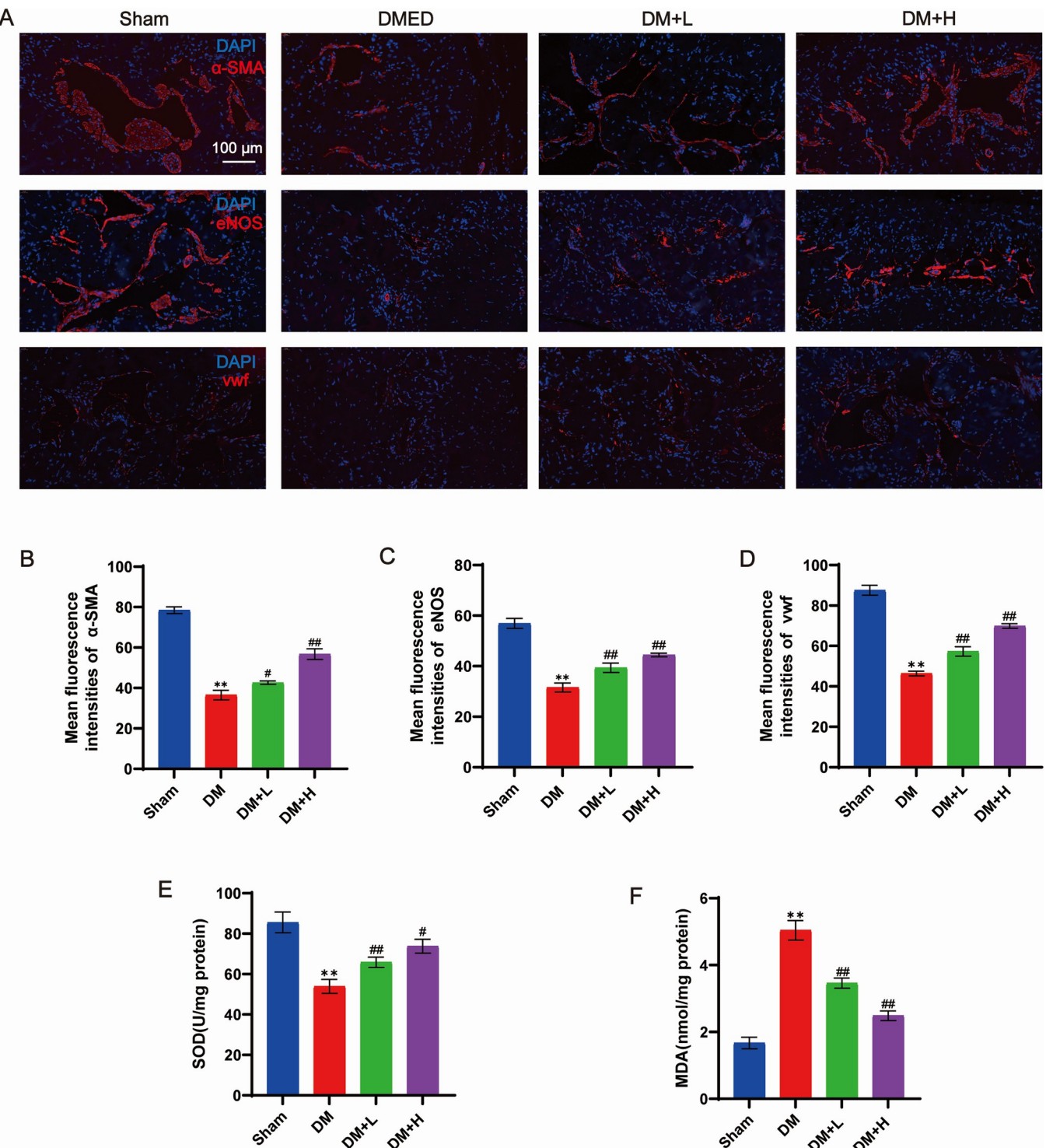

**Fig 3. Immunofluorescence of eNOS, α-SMA and vWF expression in the penile corpus cavernosum tissues of rats between groups and detection of SOD activity and MDA levels.** (A) Immunofluorescence staining was performed to visualize the presence of α-SMA, eNOS, and vWF in the corpus cavernosum of the sham, DMED, DMED+low-dose salidroside, and DMED+high-dose salidroside groups. (B-D) The mean fluorescence intensities of α-SMA, eNOS, and vWF were quantified. (E and F) The MDA and SOD contents of each group are depicted in a bar graph, with data presented as mean ± standard deviation. The data are presented as mean ± SD and were analysed using GraphPad Prism 9 software. Statistical comparisons between two groups were performed using an unpaired t test, while comparisons between three groups were analysed using one-way ANOVA followed by Tukey's post hoc test. **$P < 0.01$ signifies a noteworthy distinction compared to the sham group. ##$P < 0.01$ #$P < 0.05$ signifies a noteworthy distinction compared to the DM group.

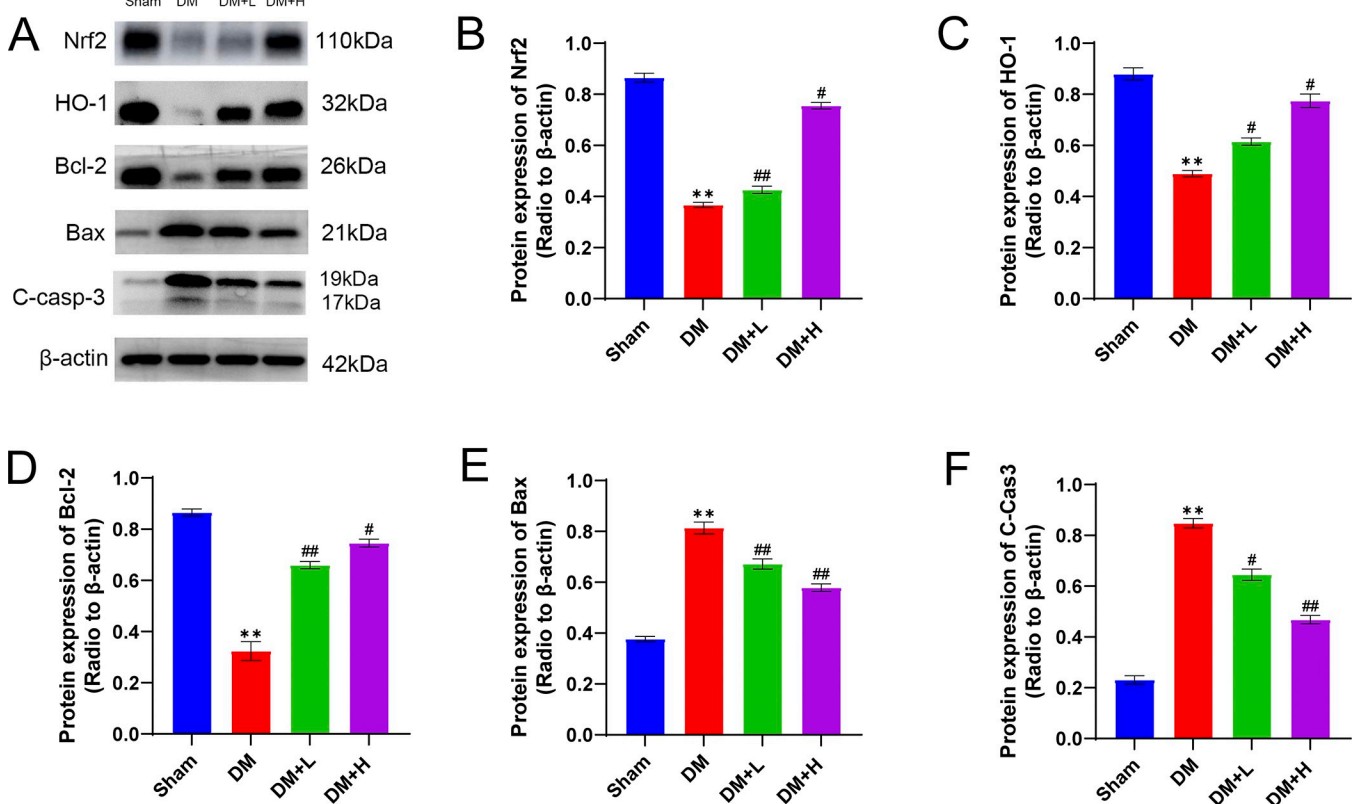

**Fig 4. Expression of Nrf2/HO-1 pathway and apoptosis-related proteins.** (A) Detection of the expression of Nrf2, HO-1, Bcl-2, Bax, and cleaved caspase-3 in rat corpus cavernosum tissue using Western blotting. (B-F) Bar graphs illustrate the expression levels of Nrf2/HO-1 pathway proteins and apoptosis-related proteins in the corpus cavernosum of rats following different treatments. The data are presented in the form of the comparative intensity in relation to β-actin. (Tukey's multiple comparisons test, one-way ANOVA, **P<0.01, #P<0.05, ##P<0.01.) Data are displayed as mean ± SD and were analysed by GraphPad Prism 9 software.

H2O2 in comparison to the control group (P<0.05, Fig 5C). Prior exposure to 40 μg/ml salidroside mitigated the adverse impacts of H2O2 on the viability of CCSMCs (in comparison to the H2O2 group, P<0.05, Fig 5D), as evidenced by the CCK-8 results.

## Effect of salidroside pretreatment on ROS in CCSMCs under H2O2 intervention

By flow cytometry analysis, hydrogen peroxide treatment markedly elevated the levels of ROS in CCSMCs compared to controls (P<0.01, P<0.01, Fig 5E and 5F). Prior administration of salidroside significantly decreased cellular ROS (P<0.01, P<0.01, Fig 5E and 5F).

## The impact of salidroside pretreatment on the levels of protein expression in CCSMCs during H2O2 exposure

In the $H_2O_2$ group, the Western blot findings indicated the expression levels of Nrf2, HO-1, and Bcl-2 in CCSMCs were significantly decreased compared to the sham group, while there was a notable rise in the levels of Bax and cleaved caspase-3 (P<0.01, Fig 6A–6F). The salidroside pretreatment group exhibited elevated levels of Nrf2, HO-1, and Bcl-2 expression (P<0.01, Fig 6B–6D). Moreover, there was a decrease in the expression levels of Bax and cleaved caspase-3 (P<0.01, Fig 6E and 6F) when compared to the H2O2 group.

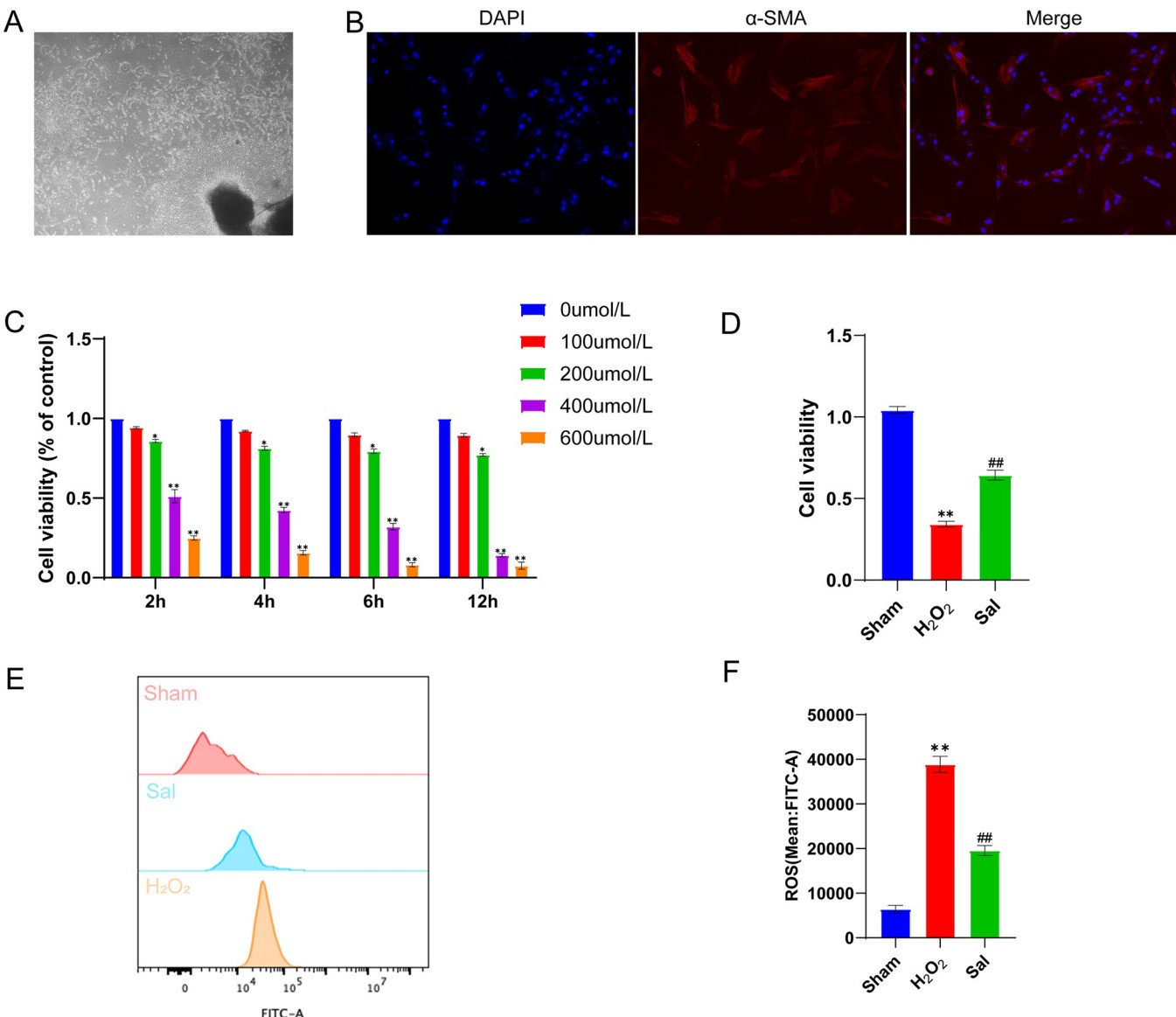

**Fig 5. Salidroside pretreatment on CCSMCs under exposure to H₂O₂.** (A) Extraction of primary CCSMCs of SD rats. (B) Immunofluorescence identification of CCSMCs (α-SMA). (C) $H_2O_2$ time–concentration gradient experiment and CCK-8 assay to evaluate the impact of interventions on the viability of CCSMCs in each group. (D) CCK-8 assay to evaluate the effects of pretreatment with salidroside on the viability of CCSMCs under $H_2O_2$ treatment. (E-F) Flow cytometry analysis to assess the effects of pretreatment with salidroside on ROS levels in CCSMCs under $H_2O_2$ treatment. Compared with the Sham group, **$P<0.01$, indicating statistical significance; compared with the H2O2 group, ## $P<0.01$, indicating statistical significance.

## Discussion

The prevalence of diabetes has been consistently rising over time, rendering it one of the prevalent chronic illnesses, distinguished by disrupted glucose metabolism. The increasing occurrence and prevalence of diabetes present a notable health issue because of the complications it is linked to. It is widely acknowledged that males suffering from diabetes face a notably elevated likelihood of encountering erectile dysfunction (ED) in comparison to their non-diabetic counterparts. This condition has a profound impact on their overall quality of life [23]. ED is a multifaceted condition characterized by an intricate etiology, encompassing various systemic

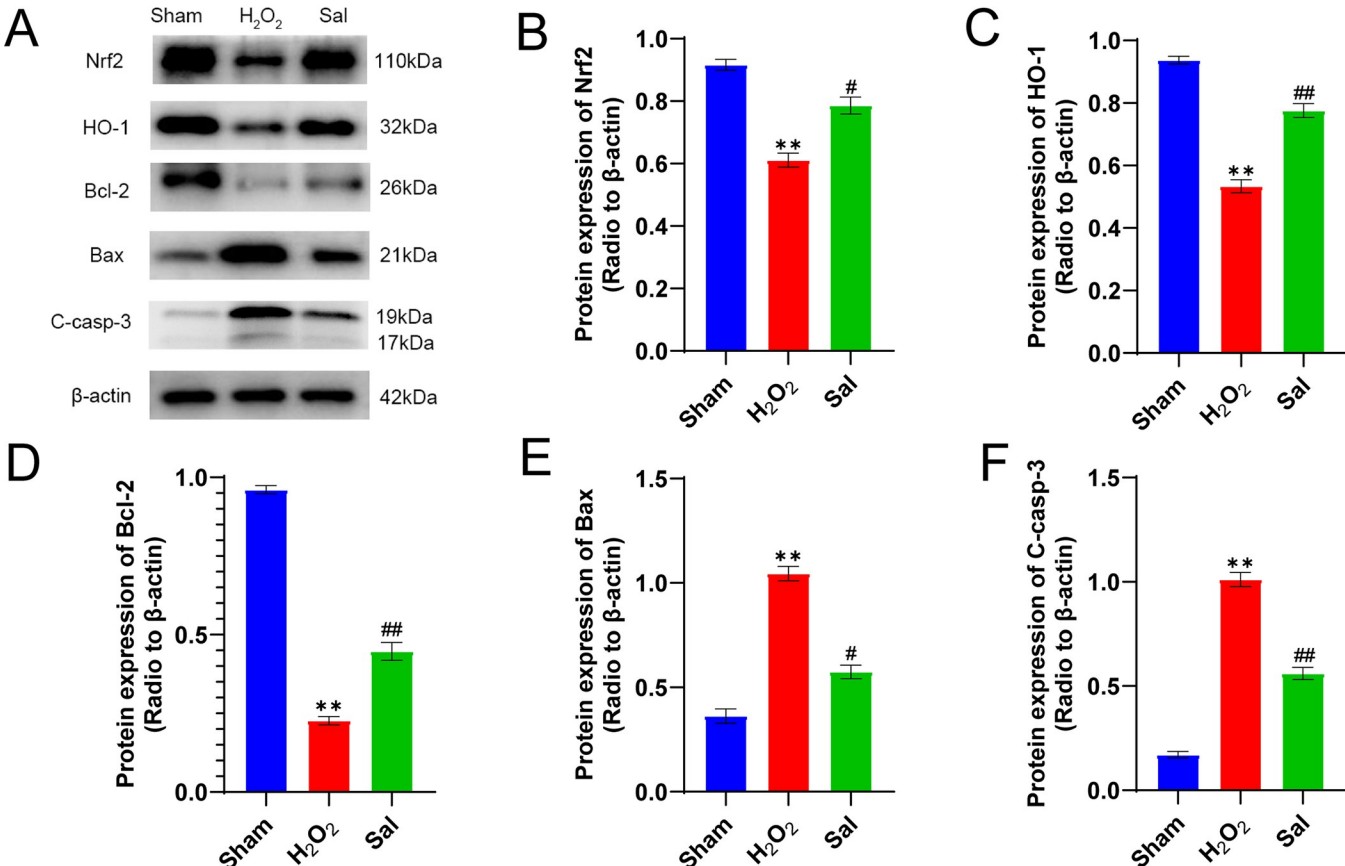

**Fig 6. Salidroside pretreatment on the expression levels of Nrf2, HO-1, Bcl-2, Bax, and cleaved caspase-3 in CCSMCs after H₂O₂ exposure.** (A) Effects of salidroside pretreatment on the expression levels of Nrf2, HO-1, Bcl-2, Bax, and cleaved caspase-3 in CCSMCs after H₂O₂ exposure. β-Actin was used as a loading control. (B and C) The expression of the Nrf2/HO-1 pathway at the protein level after different treatments was detected by WB. (D-F) Expression of apoptosis-related proteins after salidroside or H₂O₂ treatment. (Tukey's multiple comparisons test, one-way ANOVA, $**P<0.01$, $\#P<0.05$, $\#\#P<0.01$).

changes that frequently co-occur and mutually influence one another. Besides diabetes, other factors that commonly contribute to ED are advanced age, dyslipidaemia, high blood pressure, heart disease, obesity, metabolic syndrome, sedentary lifestyle, and tobacco use. In addition to low testosterone levels, other factors that contribute to the clinical scenario, such as neurogenic and iatrogenic factors (medication or surgical treatment) as well as psychological factors, further complicate the situation [24,25].

Lately, there has been an increasing emphasis on utilizing natural plant extracts derived from traditional Chinese medicine monomers because of their beneficial pharmacological properties and limited adverse reactions [26]. Some studies have explored the therapeutic impacts and mechanisms of salidroside, a key compound found in Rhodiola, a plant used for medicinal and culinary purposes [27]. Salidroside has shown remarkable anti-inflammatory and antioxidant qualities, making it suitable for treating different ailments. In this study, the primary focus was on investigating the effects of salidroside on diabetic rats experiencing erectile dysfunction. The findings strongly suggest that salidroside possesses the capacity to serve as a viable therapeutic option for erectile dysfunction resulting from diabetes, potentially owing to its wide-ranging pharmacological impacts.

First, we measured the ICP and MAP in the corpus cavernosum of rats following electrical activation of the CN. The comparison of the maximum ICP/MAP ratios revealed a significant

impairment in erectile function in DMED rats, whereas treatment with salidroside demonstrated a notable improvement in the erectile function of DMED rats. The potential therapeutic effects of salidroside on erectile dysfunction associated with diabetes are speculated to be achieved by increasing the activity of the Nrf2/HO-1 signalling pathway, which in turn reduces oxidative stress levels in the tissue of the penile corpus cavernosum, decreases cell apoptosis, and alleviates fibrosis [7].

SOD and MDA levels were measured in the penile corpus cavernosum tissue of rats with DMED. Additionally, the protein expression of Nrf2 and HO-1 was analysed in this study. The findings indicated a decline in SOD function, an elevation in MDA concentrations, and a decrease in the levels of Nrf2 and HO-1 proteins in the penile tissue of DMED rats. The results indicate that the antioxidant pathway in the DMED rats is inhibited, resulting in an aggravation of oxidative stress. We noticed that the administration of salidroside counteracted the aforementioned patterns. The results of the research indicated that salidroside had the capability to enhance the Nrf2/HO-1 signalling pathway, decrease oxidative stress, and enhance erectile function in rats with DMED. This implies that salidroside might produce its effects by activating antioxidant and anti-inflammatory processes, resulting in an improvement in erectile dysfunction.

One of the pathological features of DMED is the excessive apoptosis observed in the penile corpus cavernosum cells. Extended periods of high blood sugar can trigger cell death through multiple pathways, which encompass oxidative stress, endoplasmic reticulum stress, and inflammation [28]. The Bcl-2 gene family regulates caspase-3, which is widely acknowledged as a crucial facilitator of programmed cell death. Bcl-2 functions as a protein that inhibits apoptosis, while Bax acts as a protein that promotes apoptosis. Studies have indicated that as diabetes progresses, there is a decline in the expression of Bcl-2 and a rise in the expression of Bax, resulting in the activation of caspase-3 and the initiation of cellular apoptosis. In our investigation, we utilized Western blot analysis to identify a reduction in Bcl-2 levels and an elevation in Bax and cleaved caspase-3 levels within the penile corpus cavernosum tissue of DMED rats. Severe cell apoptosis in the penile corpus cavernosum was confirmed through TUNEL staining. The administration of salidroside mitigated these effects, indicating that salidroside might relieve cellular apoptosis triggered by DM. Additional research is necessary to clarify the precise mechanisms.

Fibrosis of the penis is a significant pathological alteration that happens during the progression of DMED and is a frequent factor contributing to treatment-resistant erectile dysfunction. Excessive release of fibrotic substances in the corpus cavernosum leads to fibrosis, causing an abnormal buildup of collagen, deposition of extracellular matrix, and decreased smooth muscle content. The use of Masson's trichrome staining and immunofluorescence allowed us to observe a notable decline in the levels of the smooth muscle marker α-SMA, the endothelial marker vWF, and endothelial nitric oxide synthase (eNOS) in the tissue of the penile corpus cavernosum in rats with DMED [29]. These suggested a decrease in smooth muscle content, damage to endothelial cells, an increase in collagen deposition, and the development of severe penile fibrosis. The administration of salidroside notably enhanced the smooth muscle, endothelium, and eNOS levels, indicating that salidroside therapy can alleviate erectile dysfunction in DMED rats by decreasing penile fibrosis.

To confirm the beneficial impact of salidroside on CCSMCs, we conducted cellular experiments by subjecting them to H2O2 to mimic conditions of oxidative stress. H2O2 can lead to additional harm to cells by inducing pathological alterations, such as oxidative stress and apoptosis. The findings indicated that CCSMCs displayed reduced cell viability, elevated levels of ROS, and more apoptosis when exposed to H2O2, along with decreased levels of Nrf2, HO-1, and Bcl-2 proteins and increased levels of Bax and cleaved caspase-3. Prior administration of salidroside effectively mitigated the influence of H2O2 on CCSMCs [30].

In summary, our research shows that salidroside can enhance erectile function in rats with DMED, potentially by increasing the activity of the Nrf2/HO-1 signalling pathway, decreasing oxidative stress, and alleviating cell apoptosis. Nevertheless, the intricate mechanisms of oxidative stress and cell apoptosis in DMED must be comprehensively explored to determine whether salidroside extract influences other pathways.

Our study has some limitations, such as that the animal model used only represents type 1 DMED. Further experimental research is necessary to explore the more aetiology in more depth.

## Conclusion

Salidroside mitigates oxidative stress and decrease apoptosis in the corpus cavernosum tissue of male rats with diabetes. This enhances erectile function and could be used as a treatment for diabetic erectile dysfunction (DMED). Moreover, prior application of salidroside mitigates the harmful impacts of hydrogen peroxide stimulation on the viability of CCSMCs, leading to a decrease in the production of reactive oxygen species (ROS) and a reduction in cellular apoptosis. The results emphasize the value of focusing on the pathways of oxidative stress and apoptosis as a hopeful strategy for managing DMED.

## Supporting information

**S1 Fig. Three sets of ICP/MAP bar graph data.**
(XLS)

**S2 Fig. Bar graph data on the ratio of collagen to smooth muscle in the corpus cavernosum.**
(XLS)

**S3 Fig. Average fluorescence intensity of TUNEL assay.**
(XLS)

**S4 Fig. The mean fluorescence intensities of α-SMA, eNOS, and vWF were quantified.**
(XLS)

**S5 Fig. The MDA and SOD contents of each group are depicted in a bar graph.**
(XLS)

**S6 Fig. Expression of Nrf2/HO-1 pathway and apoptosis-related proteins.**
(XLS)

**S7 Fig. Salidroside pretreatment on CCSMCs under exposure to H2O2.**
(XLS)

**S8 Fig. Salidroside pretreatment on the expression levels of Nrf2, HO-1, Bcl-2, Bax, and cleaved caspase-3 in CCSMCs after H2O2 exposure.**
(XLS)

**S1 Raw images.**
(PDF)

## Acknowledgments

We thank the member of Engineering Laboratory of Urinary Organ and Functional Reconstruction of Shandong Province for their discussion and guidance on this study.

## Author Contributions

**Conceptualization:** Qiang Fu.

**Investigation:** Zhenghao Li, Zhongkai Guo, Keqin Zhang, Ziheng Li.

**Supervision:** Danfeng Zhao, Qiang Fu.

**Validation:** Zhenghao Li, Bin Jia.

**Writing – original draft:** Zhenghao Li.

**Writing – review & editing:** Qiang Fu.

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
