## [Decision Letter · Decision Letter 0]

1 Mar 2024

PONE-D-24-00064Therapeutic Potential of Salidroside in Diabetic Erectile Dysfunction: Attenuation of Oxidative Stress and Apoptosis via the Nrf2/HO-1 PathwayPLOS ONE

Dear Dr. FU,

Thank you for submitting your manuscript to PLOS ONE. After careful consideration, we feel that it has merit but does not fully meet PLOS ONE’s publication criteria as it currently stands. Therefore, we invite you to submit a revised version of the manuscript that addresses the points raised during the review process.

We look forward to receiving your revised manuscript.

Kind regards,

Md. Jamal Uddin

Academic Editor

PLOS ONE

Journal Requirements:

 "Funding for this study was provided by the National Natural Science Foundation of China under Grant No. 82071635, the Jinan New Colleges and University under Grant No. 2021GXRC085, and the Academic Promotion Programme (2020LI001) of  Shandong First Medical University."

 "Funding for this study was provided by the National Natural Science Foundation of China under Grant No. 82071635, the Jinan New Colleges and University under Grant No. 2021GXRC085, and the Academic Promotion Programme (2020LI001) of  Shandong First Medical University."

"The author(s) received no specific funding for this work"

7. We note that your Data Availability Statement is currently as follows:All relevant data are within the manuscript and its Supporting Information files.

8. Your ethics statement should only appear in the Methods section of your manuscript. If your ethics statement is written in any section besides the Methods, please move it to the Methods section and delete it from any other section. Please ensure that your ethics statement is included in your manuscript, as the ethics statement entered into the online submission form will not be published alongside your manuscript.

9. Please include a copy of Table 1 which you refer to in your text on page 14.

10. Please include captions for your Supporting Information files at the end of your manuscript, and update any in-text citations to match accordingly. Please see our Supporting Information guidelines for more information: http://journals.plos.org/plosone/s/supporting-information.

Reviewers' comments:

Reviewer's Responses to Questions

**Comments to the Author**

1. Is the manuscript technically sound, and do the data support the conclusions?

Reviewer #1: Yes

Reviewer #2: Yes

2. Has the statistical analysis been performed appropriately and rigorously? 

Reviewer #1: Yes

Reviewer #2: Yes

3. Have the authors made all data underlying the findings in their manuscript fully available?

Reviewer #1: Yes

Reviewer #2: Yes

4. Is the manuscript presented in an intelligible fashion and written in standard English?

Reviewer #1: Yes

Reviewer #2: Yes

5. Review Comments to the Author

Reviewer #1: Comments to authors:

I am very much delighted to review the manuscript (PONE-D-24-00064) and found that the authors have put their valuable efforts and knowledge to do an innovative research and have prepared the manuscript nicely.

The manuscript has been written following clear English language and grammar. The Introduction chapter contains relevant background literature and well-referenced information. All the figures, table and data have been provided which are statistically measured and presented.

Experimental design and methods are well described. The analytical parameters are adequate for the interpretation and discussion of the results. Research questions are well-defined and explored. Results indicate that salidroside upregulated the Nrf2/HO-1 signalling pathway, leading to the improvement of erectile function in diabetic male rats by alleviating oxidative stress and reducing apoptosis in corpus cavernosum tissue. This is an original research paper so far. Preprint of the same article is available at https://doi.org/10.21203/rs.3.rs-3446173/v1.

The conclusion summarizes findings, and discusses the implications. Over all, the manuscript can be accepted for publication.

Reviewer #2: Authors of the manuscript (PONE-D-24-00064) determines the therapeutic potential of Salidroside to ameliorate experimentally induced diabetic erectile dysfunction in SD rats. It’s an interesting study and the results of the study might be valuable to find out effective therapeutic intervention to reduce or overcome DMED. The design of the experiment is acceptable and overall data presentation and write up of the manuscript is good. However, authors may follow the comments below to improve it further:

1. Authors studied the therapeutic effects of Salidroside in type-1 DMED. Though, they mentioned their study lamination in discussion section, it might be great to rephrase the title accordingly to make the title more specific.

2. It would be great if protein level data were supported by up or down regulation of specific genes determined by qRT-PCR.

3. In concluding remarks author’s mention that “The activation of the Nrf2 pathway, which is essential for cellular antioxidant defence mechanisms, is probably responsible for this outcome”. This statement is not based on their findings. Authors may exclude this statement from concluding remarks and discuss their hypothesis in discussion section as future direction/recommendation.

6. PLOS authors have the option to publish the peer review history of their article (what does this mean?). If published, this will include your full peer review and any attached files.

Reviewer #1: No

Reviewer #2: **Yes: **Dr. Md. Masudur Rahman

---

## [Author Response · Author response to Decision Letter 0]

7 Jun 2024

Dear Reviewers,

We would like to express our sincere gratitude for your insightful and constructive feedback on our manuscript titled "[Manuscript Title]" (PONE-D-24-00064). Your comments have been invaluable in enhancing the quality and clarity of our work, and we truly appreciate the time and effort you have dedicated to reviewing our manuscript.

To Reviewer #1:

We are delighted that you found our research innovative and well-prepared. Your positive assessment of the clarity of language, adequacy of experimental design and methods, and the relevance of our findings to the field is truly encouraging. We are grateful for your acknowledgment of the originality of our research and your recognition of the significance of our findings. We have duly noted your comments regarding the availability of the preprint version of our manuscript and will ensure that any necessary updates or revisions are reflected in the final published version.

To Reviewer Dr. Md. Masudur Rahman:

We appreciate your thoughtful comments and suggestions for further improvement of our manuscript. We agree with your suggestion to consider rephrasing the title to specify the type of diabetic erectile dysfunction (DMED) studied in our research. In addition, we believe that the results of Western blot demonstrate a change regarding the level of the Nrf2 pathway from the protein expression level, which is more intuitive than qRT-PCR analysis of the gene. Regarding the concluding remarks, we acknowledge that the statement about activation of the Nrf2 pathway should be supported by further evidence and will address this issue accordingly in the revised manuscript.

In response to your feedback, we have already revising the manuscript to address the specific points raised by both reviewers. We are confident that these revisions will enhance the clarity, specificity, and overall quality of the manuscript.

Once again, we extend our sincere appreciation for your valuable feedback and constructive criticism. We are committed to addressing all concerns and ensuring that our manuscript meets the standards of PLOS ONE. We look forward to submitting the revised version for your consideration.

Thank you for your continued support and guidance throughout this review process.

Best regards,

Zhenghao Li

---

## [Editor Report · Decision Letter 1]

25 Jun 2024

Therapeutic Potential of Salidroside in Type ⅠDiabetic Erectile Dysfunction: Attenuation of Oxidative Stress and Apoptosis via the Nrf2/HO-1 Pathway

PONE-D-24-00064R1

Dear Dr. FU,

We’re pleased to inform you that your manuscript has been judged scientifically suitable for publication and will be formally accepted for publication once it meets all outstanding technical requirements.

Kind regards,

Md. Jamal Uddin

Academic Editor

PLOS ONE

---

## [Editor Report · Acceptance letter]

3 Jul 2024

PONE-D-24-00064R1 

PLOS ONE

Dear Dr. Fu, 

I'm pleased to inform you that your manuscript has been deemed suitable for publication in PLOS ONE. Congratulations! Your manuscript is now being handed over to our production team.

Kind regards, 

on behalf of

Dr. Md. Jamal Uddin 

Academic Editor

PLOS ONE